# Distribution and Quantification of Lactic Acid Enantiomers in Baijiu

**DOI:** 10.3390/foods11172607

**Published:** 2022-08-27

**Authors:** Hao Xu, Shuyi Qiu, Yifeng Dai, Yuangen Wu, Xiangyong Zeng

**Affiliations:** 1Guizhou Province Key Laboratory of Fermentation Engineering and Biopharmacy, School of Liquor and Food Engineering, Guizhou University, Guiyang 550025, China; 2Beijing Advanced Innovation Center for Food Nutrition and Human Health, Beijing Technology and Business University, Beijing 100048, China

**Keywords:** D-lactic acid, L-lactic acid, Baijiu, chiral, HPLC, threshold, TAVs

## Abstract

Enantiomers of lactic acid were investigated in Baijiu, including soy sauce aroma-type Baijiu (SSB), strong aroma-type Baijiu (STB), and light aroma-type Baijiu (LTB), via high-performance liquid chromatography with a chiral separation column. The natural concentration and enantiomeric distribution of lactic acid were studied, and their contribution to the flavor of Chinese Baijiu was evaluated based on recognition threshold. The results showed that there were significant differences in the content of lactic acid and the ratio of enantiomeric isomers among different aroma types and storage year. In SSB, the concentrations of D-lactic acid and L-lactic acid were higher, with the highest concentrations of 1985.58 ± 11.34 mg/L and 975.31 ± 14.03 mg/L, respectively. In STB, the highest concentrations of D-lactic acid and L-lactic acid were 1048.00 ± 11.46 mg/L and 939.83 ± 0.23 mg/L, respectively. In LTB, the highest concentrations of D-lactic acid and L-lactic acid were 760.90 ± 9.45 mg/L and 558.33 ± 3.06 mg/L, respectively. The average D/L enantiomeric ratios were 78:22 ± 16.16 and 80:20 ± 9.72 in the Commercial Baijiu products of SSB and STB, respectively. The average D/L enantiomeric ratio in LTB was 90:10 ± 6.08. D-lactic acid in JSHS vintage Baijiu showed a wave variation with aging, while L-lactic acid gradually increased during aging, and the average D/L enantiomeric ratio was 76:24 ± 4.26. The concentration of D-lactic acid in XJCT vintage Baijiu also showed a wave variation with aging, and the concentration of L-lactic acid tended to be stable during aging, with an average D/L enantiomeric ratio of 88:12 ± 2.80. The content of the two configurations of lactic acid in the LZLJ vintage Baijiu showed a decreasing trend during aging, with an average D/L enantiomeric ratio of 60:40 ± 11.99. The recognition threshold of D-lactic acid in 46% ethanol solution was 194.18 mg/L with sour taste; while the L-lactic acid was 98.19 mg/L with sour taste. The recognition threshold of L-lactic acid was about half that of D-lactic acid, indicating that L-lactic acid has a stronger sour taste. The taste activity values (TAVs) of D-lactic acid and L-lactic acid were greater than 1 in most of the Baijiu samples, and the TAV of D-lactic acid was greater than that of L-lactic acid. The study showed that the lactic acid enantiomers contributed to the taste perception of Baijiu in most of the samples, and D-lactic acid contributed more to the Baijiu taste than L-lactic acid.

## 1. Introduction

Baijiu, also called shao jiu, is a world-renowned distilled alcoholic beverage with a long history that occupies an indispensable position in China. It is produced by a unique brewing method, and is an important contributor to the global beverage alcohol industry [1,2]. Solid-state Baijiu brewing is mainly based on sorghum, with Daqu being used as a saccharification starter, and Baijiu from different regions has significant differences in aroma characteristics due to different raw materials and specialized brewing techniques [3]. Based on flavor types, Baijiu is divided into 12 aroma types, of which SSB, STB, and LTB dominate the market; the three types account for more than 70% of the Chinese Baijiu market [4,5]. Studies on the differentiation of different aroma types of Baijiu and their authenticity usually involve sensory evaluation, as well as some characteristic chemical compounds. There are certain differences in the content and flavor of certain characteristic chiral flavor substances in different aromatic Baijiu, such as the 1,2-propanediol isomers in Baijiu that we studied previously, where the S-configuration was dominant in SSB, the R-enantiomer was dominant in STB, while neither S- nor R-enantiomer were detected in LTB [6].

Lactic acid has a wide range of applications in the food, cosmetic, agricultural, and pharmaceutical industries. The naturally occurring lactic acid in nature is L-lactic acid, while D-lactic acid is mainly produced by synthesis [7,8]. As reported, human has only L-Lactic acid dehydrogenase to metabolize L-lactic acid and can only absorb L-lactic acid, and elevated levels of the D-lactic acid are harmful, so L-lactic acid is the preferred isomer in the food and pharmaceutical industry [9,10]. Lactic acid is a chiral metabolite and the two enantiomeric isomers have different biochemical properties, and variations in the single enantiomeric form have been found to be strongly associated with certain diseases [11].

Lactic acid is described as being slightly acidic, sweet, astringent, and thick, which can stabilize the aroma of the Baijiu, and also reduce the bitterness and astringency in Baijiu, making the Baijiu mellow [12]. Lactic acid is an important flavoring and buffering substance in Baijiu, as well as a precursor of ethyl lactate [13,14]. In STB, lactic acid has a good aromatic aid to the main aroma component ethyl caproate, which plays a role in buffering and harmonizing the Baijiu flavor, masking the irritation of alcohol, and also possessing affinity with a variety of components to make the Baijiu more harmonious [15,16]. Lactic acid was detected in all samples of rice-flavored Baijiu, and it was mainly produced by *Aspergillus oryzae* and lactic acid bacteria. The correlation between lactic acid levels and total acidity levels in rice-flavored Baijiu indicates that high lactic acid levels lead to a higher total acidity in rice-flavored Baijiu [17]. In addition, lactic acid was verified to play an important role in the roasted sesame-flavored Baijiu by omission experiments [18]. Lactic acid is an important organic acid in Japanese sake and wine and is thought to have a strong influence on the flavor [19]. D-lactic acid and L-lactic acid are also widely studied components of wine, and their relative concentrations in a given wine depend on the microbial history of the wine [20]. D-lactic acid is present in wines but not found in grape juice, and L-lactic acid is only significantly present in wines that have undergone malolactic-lactic fermentation [21]. Jiang et al. [22] determined both enantiomers of lactic acid in Baijiu, and the results showed that the D-configuration was significantly higher than L-lactic acid in some Baijiu samples studied. Liu et al. [23] separated and measured L-lactic acid and D-lactic acid in kimchi and concluded that L-lactic acid was dominant in kimchi, and the content of L-lactic acid gradually increased during the fermentation process, while the content of D-lactic acid first increased and then decreased. Nanjo et al. [24] developed an enzyme sensor system for the simultaneous determination of D-lactic acid and L-lactic acid in alcoholic beverages. The olfactory threshold and aroma characteristics of chiral flavor compounds may vary depending on the stereoisomers considered, so lactic acid isomers must be isolated to obtain an accurate assessment of their flavor characteristics.

The molecular structure of lactic acid has an asymmetric chiral carbon atom and there are two different enantiomers (Figure 1). We collected some methods for the determination of lactic acid in Baijiu in our previous work, among which Jiang [22] et al. studied the enantiomers of lactic acid in Baijiu, but the scope of their study was small and the data were not systematically organized, and the distribution and content of its two isomers and their impact on the sensory effects of the Baijiu were not clear. Therefore, the aim of this work was to isolate and determine D-lactic acid and L-lactic acid in SSB, STB, and LTB and the corresponding vintages to explore whether the enantiomeric distribution and content of the two isomers of lactic acid in different Baijiu are related to the aroma-type and storage year, and to evaluate their sensory effects on Baijiu, respectively.

## 2. Materials and Methods

### 2.1. Chemicals

The following chemicals were used: L-lactic acid, chromatographic pure, 98%, (Beijing Solabao Technology Co., Ltd., Beijing, China); D-lactic acid, chromatographic pure, 93%, (Shanghai Maclean Technology Co., Ltd., Shanghai, China); Isopropyl alcohol, (Tianjin Comio Chemical Reagent Co., Tianjin, China); Copper sulfate-pentahydrate (CuSO_4_ 5H_2_O), 99%, (Shanghai Maclean Technology Co., Shanghai, China).

### 2.2. Samples

The determination of D-and L-lactic acid was carried out in 46 types of soy sauce aroma-type Baijiu, 17 types of strong aroma-type Baijiu, and 6 types of light aroma-type Baijiu. In this paper, the 69 samples were divided into 2 categories, in which soy sauce aroma-type Baijiu included commercially Baijiu products, JSHSJ vintage Baijiu (5–41 aging years), XJCTJ vintage Baijiu (1–11 aging years); strong aroma-type Baijiu included commercially available Baijiu products and LZLJ vintage Baijiu (2–9 aging years); and light aroma-type Baijiu included commercially available Baijiu products.

### 2.3. Sample Preparation

We took 1 mL of Baijiu sample into a 10 mL centrifuge tube, added 4 mL of ultrapure water, vortexed to mix, and then used 0.22 μm aqueous microporous membrane to filter through a BLC-1 glass sand core filter device for high-performance liquid chromatography determination.

### 2.4. Isolation and Quantification of Lactic Acid Enantiomers by High Performance Liquid Chromatography

The two isomers of lactic acid were determined using high-performance liquid chromatography (Agilent 1260 Series) (Agilent Technologies Ltd., Santa Clara, CA, USA). The column was Chirex 3126 (D)-penicillamine (250 mm × 4.6 mm, 5 µm, American Philomel, Guangzhou Philomena Scientific Instrument Co., Ltd., Guangzhou, Guangdong). Mobile phase: 2 mmol/L of 5% isopropanol mobile phase solution was prepared by weighing CuSO_4_·5H_2_O 0.500 g, adding 50 mL of double-distilled water to dissolve, transferring to a 100 mL volumetric flask, adding 50 mL of chromatographically pure isopropanol, and then fixing the volume with double-distilled water to 1000 mL, and ultrafiltrating under vacuum using a synthetic cellulose filter membrane with 0.45 μm pore size. The mobile phase flow rate was 1 mL/min; the column temperature was 30 °C; the detection wave length was 254 nm with a diode array detector; the standard and sample solutions were filtered through a BLC-1 glass sand core filter device with using 0.22 µm membrane before use; the injection volume was 5 µL [22,23]. The standard curves were prepared by accurately weighing a certain amount of L-lactic acid and D-lactic acid standards, respectively, and dissolving them in a 100 mL volumetric flask with ultrapure water to prepare L-lactic acid and D-lactic acid standards, and then diluting them to 0.020 mg/mL, 0.050 mg/mL, 0.100 mg/mL, 0.200 mg/mL, 0.400 mg/mL, 0.600 mg/mL, and 0.800 mg/mL, respectively. The standard solutions were filtered through a BLC-1 glass sand core filter device using a 0.22 μm microporous membrane before loading on the machine. Qualitative and quantitative analysis: the chromatograms of the samples under the same chromatographic conditions were compared with that of the lactic acid standard solution to determine the elution order of L-lactic acid and D-lactic acid in the samples according to the retention time. The content of L-lactic acid and D-lactic acid in the samples was calculated by quantification with an external standard method.

### 2.5. Recognition Thresholds

Ten trained evaluators aged 22–29 (five males and five females) were selected for this experiment. Recognition thresholds of D-and L-lactic acid were determined by a sensory panel in 46% ethanol solution, referring to the GB/T 22366-2008 sensory analysis methodology adopted (general guidelines for the determination of olfactory and flavor perception thresholds by the three-point option method (3-AFC)) and GB/T33406-2016 (Baijiu flavor GB/T33406-2016 (guidelines for determination of flavor thresholds of Baijiu)) [25,26].

3-AFC: Providing six groups of samples to each group member, each group included two blanks and a solution of the compound to be tested of known concentration. Each group of samples was randomly placed, and each sample was numbered with 3 random numbers.

All samples were poured into clean tulip Baijiu glasses respectively and judged by assessors. The evaluators kept the inlet volume of the tested sample at 0.5~2.0 mL, and rotated the sample in their mouths for about 3~5 s, so that the sample was evenly distributed in the oral cavity, and finally spat out the sample. After tasting one sample, the evaluators might gargle with water and take a break before tasting another sample. During testing, tasters were asked to identify samples that differed from the blank sample, and the reference sample was intended to allow panelists to correctly perceive the properties of each tested compound. All trials were performed 3 times to ensure the accuracy of the results [27,28].

The best estimate threshold (BET) for each rater was calculated based on the geometric mean of the highest concentration sample that was misidentified and the adjacent higher concentration sample. T_BETi_ is the individual recognition threshold; A_x_ is the highest concentration sample misidentified among raters; A_x+1_ is the concentration of the higher-order sample; T_BET_ is the group threshold.
$$T_{\mathrm{BETi}}=\sqrt{A_{x}\times A_{x+1}}$$
$$T_{\mathrm{BET}}=\sqrt[{n}]{T_{\mathrm{BTE}1}\times T_{\mathrm{BET}2}\ldots \times T_{\mathrm{BETn}}}$$

### 2.6. Statistical Analysis

Data were analyzed with to the Microsoft office excel 2018 application, box plots, PCA, and scatter plots; histograms were produced by origin2018 64 Bit, and the heat map was made by Tbtools. SPSS software (IBM SPSS Statistics 26.lnk) *t*-test was used to analyze the significant differences in lactic acid levels in different aromatic Baijiu. The level of statistical significance was 5%, *p* < 0.05. The results of the sensory tests were statistically calculated according to the national standard specifications.

## 3. Results and Discussion

### 3.1. Identification and Quantification of Lactic Acid Enantiomers in Chinese Baijiu by High Performance Liquid Chromatography

The enantiomers of lactic acid were separated using a chiral column and high-performance liquid chromatography (Figure 2a). Chromatograms of representative Baijiu samples of different aroma types showed different enantiomeric distributions (Figure 2b–d). (a) was the lactic acid standard; (b) was one of the SSB (soy sauce aroma-type Baijiu); (c) was one of the STB (strong aroma-type Baijiu); (d) was one of the LTB (light aroma-type Baijiu). Calibration curves, detection limits, recoveries, and reproducibility were used to validate the method. The linear ranges of the D- and L- configurations were 20–800 mg/L. The R^2^ values of the standard curves were greater than 0.9980, which satisfied the analytical requirements. The recoveries of the D- and L-lactic acid enantiomers standards were 94.78–106.99% and 94.16–101.90%, respectively, with the relative standard deviations (RSDs) ranging from 0.53% to 1.45% and 2.03–4.16%, respectively, and the method was reproducible. The detection limit of D-and L-lactic acid chiral isomer were 1.594 mg/L, 1.188 mg/L, respectively (Table 1). It showed that this method has a low detection limit for lactic acid enantiomers and can be applied to the detection of lactic acid enantiomers.

### 3.2. Comparison of Lactic Acid Enantiomers in Different Aromatic Baijiu

Solid state fermentation of SSB was performed using high-temperature Daqu (60–70 °C) followed by fermentation at 40–50 °C, whereas the fermentation process of STB involved the use of medium-temperature Daqu (55–60 °C) and fermentation at a temperature of 29–35 °C. In contrast, the fermentation process of LTB used (40–50 °C) low-temperature Daqu with solid-state fermentation temperatures between 28–32 °C [29,30,31]. The concentration of lactic acid isomers in SSB was higher than the other two flavor Baijiu, which may be because the temperature of Daqu production and fermentation was high-temperature solid-state fermentation, which was conducive to the metabolic degradation of microorganisms. D-lactic acid accounted for more than 50% of the lactic acid enantiomers in 98% of the 69 Baijiu samples, which was the dominant configuration. Therefore, if the L-lactic acid content in the Baijiu was predominantly enantiomeric, there might be a suspicion of adulteration (Table 2). The average concentration of D-lactic acid and L-lactic acid in Commercial products of SSB was higher than that of STB and LTB. The average concentrations of D-lactic acid in the Commercial SSB products, Commercial STB products, and Commercial LTB products were 1300.18 ± 447.40 mg/L, 495.84 ± 155.38 mg/L, and 453.17 ± 260.05 mg/L, respectively. The average concentration of L-lactic acid in soy sauce aroma-type Commercial Baijiu products, strong aroma-type Commercial Baijiu products and Light aroma-type Commercial Baijiu products were 397.54 ± 289.81 mg/L, 151.40 ± 98.26 mg/L, and 143.07 ± 206.94 mg/L, respectively (Table 3). In the JSHSJ vintage Baijiu, there were no significant changes in the enantiomeric ratio. While, in the LZLJ Vintage Baijiu of STB, the contents of D- and L-lactic acid were increased at the same time and led to the ratio of D/L decreasing to 60:40 ± 11.99.

It was found that lactic acid existed mainly in the D-configuration in SSB; the concentrations of D-lactic acid and L-lactic acid were significantly different (*p* < 0.001) (Figure 3a). There was no significant difference in the concentrations of D-lactic acid and L-lactic acid in STB (*p* > 0.05) (Figure 3b). The concentration of D-lactic acid was found to be significantly higher (*p* < 0.001) than L-lactic acid in the studied LTB, with a significant predominance of D-lactic acid (Figure 3c). The overall comparison of the concentrations of lactic acid isomers (D-lactic acid and L-lactic acid) of LTB, STB, and SSB revealed that the D-lactic acid of SSB was significantly higher than that of STB and LTB (*p* < 0.001); the L-lactic acid of SSB was significantly higher than that of LTB (*p* < 0.001), but it was not significantly different from that of STB (*p* > 0.05) (Figure 4a,b).

So far, the formation pathway of lactic acid enantiomers in Baijiu has not been clarified. Many microorganisms have been reported to have the ability to produce L-lactic acid, such as *Lactobacillus*, *Bacillus coagulans*, and various transgenic strains [32]. Lactic acid is an important metabolite produced by lactic acid bacteria during fermentation, and high levels of lactic acid may lead to rancidity [33]. Lactic acid is a ubiquitous metabolite that occurs frequently during fermentation of fermented foods and beverages [34,35], and is one of the important organic acids produced by microbial fermentation [36]. For example, the content of lactic acid in the organic acids formed during the fermentation of Maotai-flavored Baijiu was the highest, reaching 36.20 ± 6.20 g/kg [37]. Zhao et al. [38] studied the main aroma of rice-flavored Baijiu and Japanese awamori. The lactic acid in the fermentation broth of rice-flavored Baijiu was mainly L-configuration, and its concentration gradually decreased during the fermentation process. In the awamori fermentation broth, the D-lactic acid content was higher than the L-lactic acid content. The study concluded that L-lactic acid was mainly produced during solid-state saccharification and was used to produce ethyl lactate during fermentation. It has been reported that *Rhizopus oryzae* produces almost optically pure L-lactic acid directly from starch [39]. D-lactic acid is usually produced by lactic acid bacteria such as *Lactobacillus* [40]. Kodama et al. [19] used capillary electrophoresis to determine the content of D- and L-lactic acid during sake brewing. During the brewing process, the ratio of L-lactic acid to total lactic acid gradually decreased. The yeast used for sake brewing (*Saccharomyces cerevisiae*) produced only D-lactic acid during fermentation.

### 3.3. Comparison of Lactic Acid Enantiomers in Different Types of Baijiu

To better understand the flavor characteristics of LTB, STB, and SSB, we evaluated the differences of D-lactic acid and L-lactic acid in the different types of Baijiu. The concentration of D-lactic acid in JSHSJ was significantly lower than that of soy sauce aroma-type Commercial Baijiu products and XJCT (*p* < 0.01), while there was no significant difference in the concentration of D-lactic acid between soy sauce aroma-type Commercial Baijiu products and XJCT (*p* > 0.05). There was no significant difference in L-lactic acid concentration between the soy sauce aroma-type Commercial Baijiu products and JSHSJ; L-lactic acid concentration in XJCT was lower than that of JSHSJ and soy sauce aroma-type Commercial Baijiu products (*p* < 0.01) (Figure 5a,b). Although the three different types of SSB were all from the soy sauce aroma-type Baijiu-producing area in the Chi shui River basin, there were some differences between them, and the differences between these substances may be related to the differences in raw materials and the production process. The concentrations of D-lactic acid and L-lactic acid in the strong aroma-type Commercial Baijiu products were significantly lower (*p* < 0.001) than that of the LZLJ vintage Baijiu (Figure 5c,d). Lactic acid, a prerequisite substance for the formation of ethyl lactate and esters are produced as a result of esterification between acids and alcohols, and lactic acid undergoes esterification with ethanol and other substances during the aging process of Baijiu, and may gradually decrease during aging.

### 3.4. Analysis of Lactic Acid Enantiomers in Vintage Baijiu

A wavy variation in D-lactic acid concentration with aging time was observed in the soy sauce aroma-type vintage Baijiu (JSHSJ), with an overall gradual increase; L-lactic acid gradually increased with aging. Lower concentrations of D-lactic acid and L-lactic acid were found in XICT-4 and XJCT-5 Baijiu samples in the soy sauce aroma-type vintage Baijiu (XJCTJ), and the rest of the vintage samples showed no significant changes in D-lactic acid and L-lactic acid during aging (Figure 6a,b). In the strong aroma-type vintage Baijiu (LZLJ), it was observed that D-lactic acid showed a wave variation, dropping to the lowest concentration in the LZLJ-5 Baijiu sample, and only the D-configuration was present in this Baijiu sample; D-lactic acid concentration gradually decreased with aging, and L-lactic acid also showed a wave variation in the vintage Baijiu, while it was found that LZLJ-5, LZLJ-8, and LZLJ-9 did not have L-lactic acid (Figure 6c).

### 3.5. Principal Component Analysis and Heatmap Analysis of Lactic Acid Enantiomers in Baijiu

To visualize the differences between the different aromatic Baijiu, principal component analysis (PCA) was performed on the lactic acid isomers in the three aromatic Baijiu. As shown in Figure 7a, the contribution of the first principal component (PC1) alone was 54.4%, the contribution of the second principal component (PC2) alone was 45.1%, and the cumulative contribution of the first and second principal components was 99.6% > 85.00%, which is representative and can basically reflect the change information of the data. In addition, based on the first principal component and second principal component, the three aromatic Baijiu were divided into four different regions, reflecting the differences of different aromatic Baijiu. STB and LTB were mainly distributed in the first, second, and third quadrants, SSB was mainly distributed in the third and fourth quadrants, SSB was somewhat distinguished from LTB and STB, and LTB and STB were not completely separated, which may be due to the fact that the Baijiu samples came from different enterprises, whose production environments, conditions, and processes differed slightly. The PCA results showed that different aromatic Baijiu had some differences. In addition, in order to understand the characteristics of the tested Baijiu samples, we used the measurements of D-lactic acid and L-lactic acid from 69 sets of observations for heatmap analysis and HCA (hierarchical cluster analysis). The relative changes in the contents of the two lactic acid isomers were reflected in the color changes in the heatmap, revealing the relationship between the contents of D-lactic acid and L-lactic acid in Baijiu and different flavor types (Figure 7b). The results showed that there were some differences in lactic acid isomers in Baijiu of different flavor types, the content of lactic acid isomers in SSB was the highest, and the concentration of lactic acid isomers in STB and LTB was lower.

### 3.6. Recognition Thresholds and Taste Activity Value Analysis of Lactic Acid Isomers

The contribution of lactic acid enantiomers to the overall flavor of Baijiu samples was determined based on their TAVs (the ratio of concentration to the taste threshold). The lactic acid enantiomers were subjected to taste threshold determination and flavor characterization by the sensory tasting group, and the results of the threshold determination were shown in Table 4. D-lactic acid had a sour taste; L-lactic acid also had a sour taste. The taste thresholds of the lactic acid enantiomers in 46% ethanol solution were determined by 3-AFC and the best estimate threshold method. The taste threshold was 194.18 mg/L for D-lactic acid and 98.19 mg/L for L-lactic acid. The taste threshold of L-lactic acid was less than that of D-lactic acid, indicating that L-lactic acid is more acidic. This result suggests that the stereochemistry of the molecules has a strong influence on their perception [41,42,43].

As shown in Table 5, the TAVs of lactic acid enantiomers varied in different Baijiu, and the lactic acid enantiomers contributed to the taste perception of Baijiu in most of the samples. The TAVs of D-lactic acid in most SSB were higher than that of STB and LTB. The TAVs of D-lactic acid in SSB were all greater than 1, and the TAV of D-lactic acid in ZJ2 was the highest, with a TAV of 11. The maximum value of D-lactic acid TAVs in STB and LTB were 5 and 4, respectively. The TAVs of L-lactic acid in most SSB and STB were higher than those of LTB. The TAV of L-lactic acid in TCSP was the highest, with a TAV of 10, LZLJ2015 had the highest L-lactic acid TAV with a value of 10, andthe TAV of L-lactic acid was the largest in FJZC1988, with TAV value of 6. The TAVs of D-lactic acid and L-lactic acid were different to some extent.

## 4. Conclusions

In conclusion, chiral odor active substances in Chinese Baijiu were successfully identified by high-performance liquid chromatography (HPLC), and these data complemented the current knowledge on lactic acid enantiomers in different aromatic Chinese Baijiu and reported the content and enantiomeric distribution of lactic acid enantiomers in 69 Chinese Baijiu. The concentrations of D-lactic acid and L-lactic acid were higher in SSB. It was noteworthy that the enantiomeric ratios in most Chinese Baijiu showed a predominance of D-lactic acid, with D-lactic acid content in JSHS vintage Baijiu varying in waves with aging time and L-lactic acid gradually increasing during aging; D-lactic acid content in XJCT vintage Baijiu varied in waves with aging and L-lactic acid tended to be stable during aging. D-lactic acid and L-lactic acid decreased in the process of changing in LZLJ vintage Baijiu. The flavor characteristics of the two lactic acid enantiomers were similar. The TAVs of D-lactic acid and L-lactic acid were larger in SSB, followed by STB, and finally by LTB. The TAVs of lactic acid enantiomers in most of the Baijiu samples were greater than 1, and the TAVs of D-lactic acid were greater than those of L-lactic acid. The D-lactic acid contributed more to the Baijiu than L-lactic acid. The findings of this study will further improve our understanding of chiral flavor substances in Baijiu, and are important for clarifying the dominant conformations of chiral flavor substances and the chiral isomers that contribute to the Baijiu, identifying various aroma types of Baijiu, and evaluating the quality of products.

## Figures and Tables

**Figure 1 foods-11-02607-f001:**
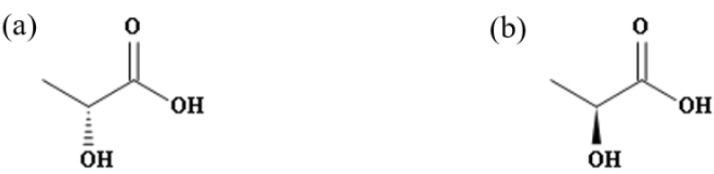
(**a**) D-lactic acid (CAS: 10326-41-7); (**b**) L-lactic acid (CAS: 79-33-4).

**Figure 2 foods-11-02607-f002:**
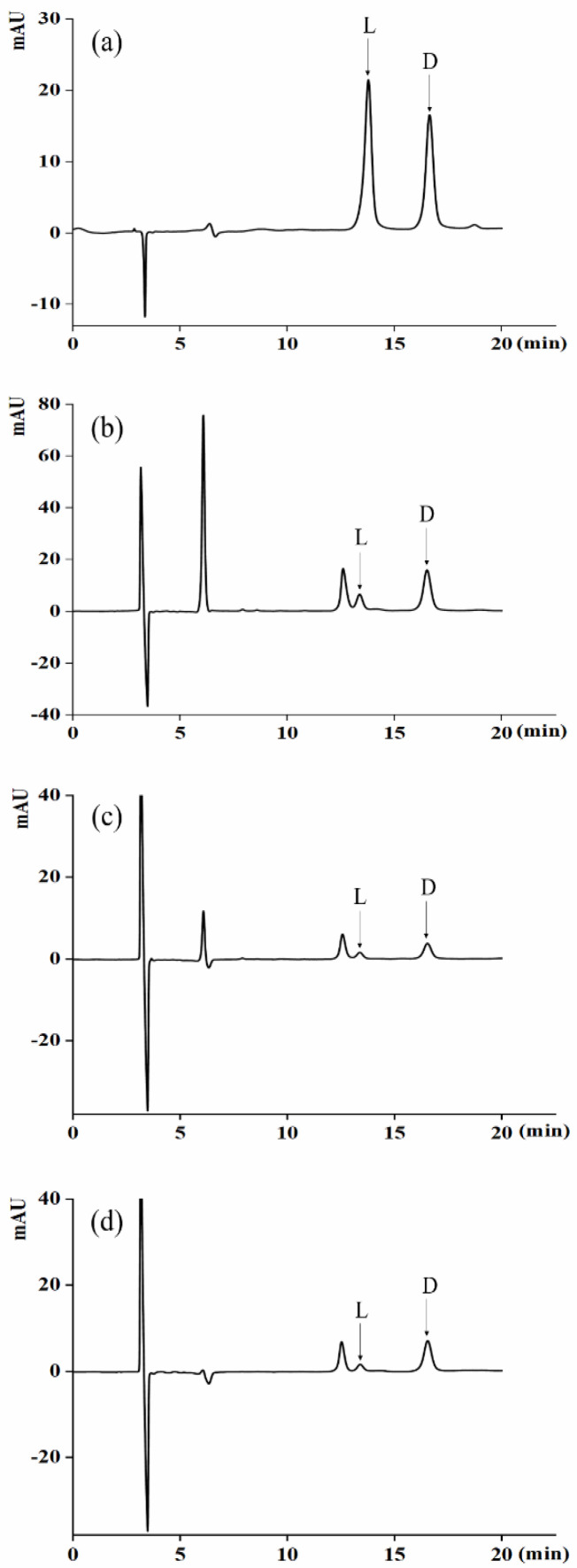
Separation of lactic acid isomers in Baijiu samples. (**a**): lactic acid standard, (**b**): MT43 Baijiu sample spectrum, (**c**): WLY Baijiu sample spectrum. (**d**): LBFJ Baijiu sample spectrum.

**Figure 3 foods-11-02607-f003:**
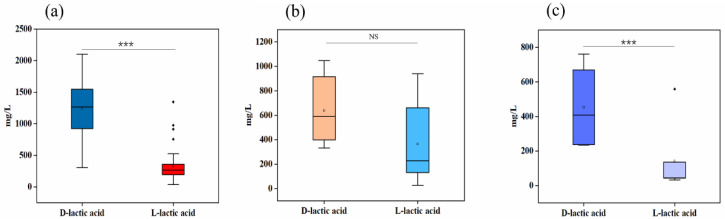
Box plots of D-lactic acid and L-lactic acid contents in SSB (**a**), STB (**b**), and LTB (**c**). *t*-test: NS (not significant, *p* > 0.05); ***: *p* < 0.001.

**Figure 4 foods-11-02607-f004:**
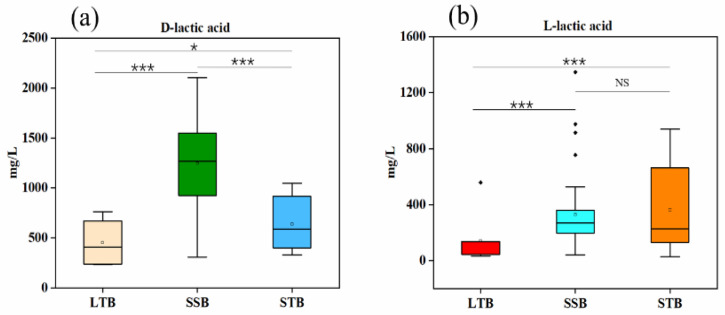
(**a**,**b**) Box plots of D-lactic acid and L-lactic acid contents in three aromatic Baijiu (SSB, STB, and LTB). *t*-test: NS (not significant, *p* > 0.05); *: 0.01 < *p* < 0.05 and ***: *p* < 0.001.

**Figure 5 foods-11-02607-f005:**
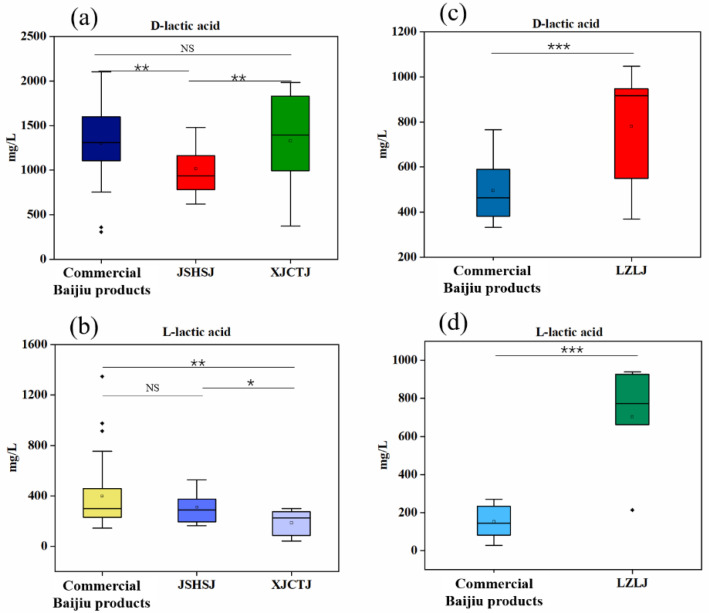
(**a**,**b**) Box plots of D-lactic acid and L-lactic acid contents in three different types of Baijiu (Commercial Baijiu products, JSHSJ, XJCTJ) in SSB; (**c**,**d**) box plots of D-lactic acid and L-lactic acid contents in two different types of Baijiu (Commercial Baijiu products, LZLJ) in STB. *t*-test: NS (not significant, *p* > 0.05); *: 0.01 < *p* < 0.05; **: 0.001 < *p* < 0.01; and ***: *p* < 0.001.

**Figure 6 foods-11-02607-f006:**
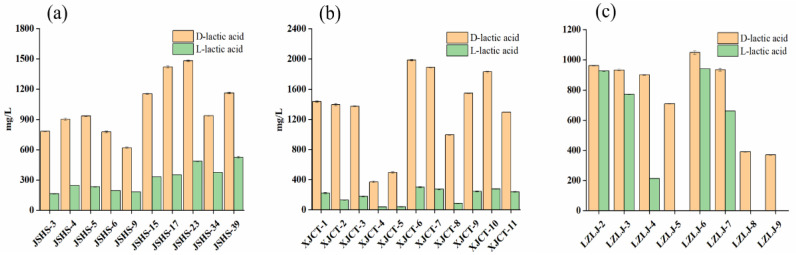
(**a**–**c**) Histograms of D-lactic acid and L-lactic acid in soy sauce aroma-type vintage Baijiu (JSHS, XJCT) and strong aroma-type vintage Baijiu (LZLJ).

**Figure 7 foods-11-02607-f007:**
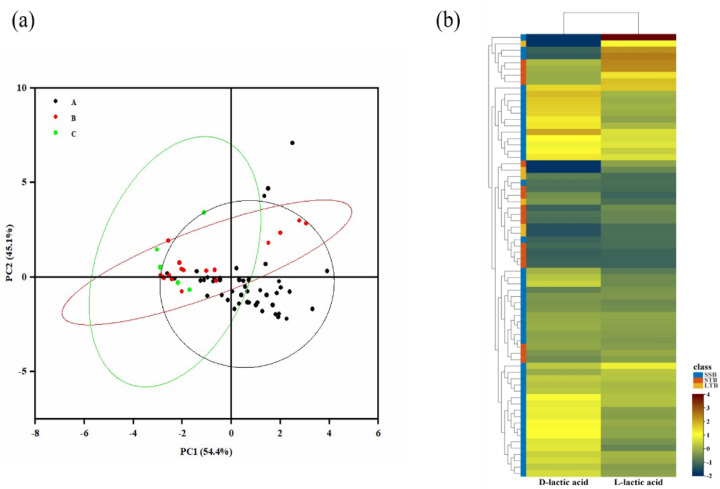
(**a**) Principal component analysis of lactic acid enantiomers in Baijiu samples, A: SSB; B: STB; C: LTB; (**b**) heatmap analysis of lactic acid enantiomers in Baijiu.

**Table 1 foods-11-02607-t001:** Correlation coefficients, LODs, linear ranges, recovery rates, RSDs.

No.	Compounds	Linearity (mg/L)	R^2^	RSD (%)	Recovery Rate (%)	LOD (mg/L)
1	D-lactic acid	20–800	0.9997	0.53–1.45%	94.78–106.99%	1.594
2	L-lactic acid	20–800	0.9985	2.03–4.16%	94.16–101.90%	1.188

Note: R^2^: Correlation coefficients, LOD: limit of detection, RSD: relative standard deviation.

**Table 2 foods-11-02607-t002:** The concentration and ratio of lactic acid enantiomers in the three major Baijiu flavors.

Sample	D-Lactic Acid (mg/L)	L-Lactic Acid (mg/L)	ee	D:L
SSB (soy sauce aroma-type Baijiu)				
LM	1545.93 ± 22.83	244.83 ± 1.62	72.66%	86:14
JSHSJ1	756.08 ± 2.39	144.92 ± 1.78	67.83%	84:16
JSHSJ2	857.13 ± 2.11	219.10 ± 3.98	59.28%	80:20
JSJ	1119.18 ± 7.23	153.90 ± 5.21	75.82%	88:12
ZJ1	1855.91 ± 3.03	295.78 ± 3.96	72.51%	86:14
ZJ2	2103.84 ± 5.79	473.05 ± 6.93	63.29%	82:18
DYT1	1173.98 ± 19.78	220.02 ± 4.37	68.43%	84:16
DYT2	1598.53 ± 4.65	405.37 ± 6.25	59.54%	80:20
GT1	1234.13 ± 9.05	303.94 ± 7.34	60.48%	80:20
GT2	1193.68 ± 7.89	359.96 ± 13.14	53.66%	77:23
GT3	1104.49 ± 4.86	318.02 ± 1.80	55.29%	78:22
XJYZ	1232.18 ± 5.16	150.60 ± 7.87	78.22%	89:11
QHL	1567.41 ± 24.85	457.44 ± 8.86	54.82%	77:23
MT43	1673.00 ± 9.16	238.07 ± 4.20	75.08%	88:12
MT53	1700.58 ± 5.81	318.85 ± 1.61	68.42%	84:16
XJ1988	1308.86 ± 2.21	287.37 ± 0.89	63.99%	82:18
QJ1H1	1525.75 ± 5.11	351.01 ± 2.72	62.59%	81:19
DYT3	1452.59 ± 8.58	229.17 ± 0.38	72.75%	86:14
MTWZJ	1529.40 ± 3.69	267.83 ± 3.66	70.20%	85:15
LJ	1803.99 ± 6.59	755.38 ± 1.17	40.97%	70:30
MTCX	359.08 ± 1.13	914.22 ± 1.01	43.60%	28:72
QJ1H2	1698.16 ± 28.78	471.82 ± 19.2	56.51%	78:22
TCSP	306.75 ± 6.11	975.31 ± 14.03	52.15%	24:76
GZJSJ1	880.63 ± 2.28	196.79 ± 4.16	64.86%	82:18
GZJSJ2	923.32 ± 5.81	236.53 ± 2.86	69.93%	85:15
JSHSJ39	1162.00 ± 7.21	525.74 ± 6.90	37.70%	69:31
JSHSJ34	936.50 ± 4.53	373.89 ± 0.32	42.93%	71:29
JSHSJ23	1480.00 ± 7.78	485.68 ± 1.41	50.58%	75:25
JSHSJ17	1420.00 ± 9.90	350.79 ± 1.47	60.38%	80:20
JSHSJ15	1153.50 ± 6.79	331.12 ± 1.09	55.39%	77:23
JSHSJ9	620.00 ± 6.36	181.60 ± 0.91	54.69%	76:24
JSHSJ6	776.50 ± 8.77	194.01 ± 0.71	60.02%	79:21
JSHSJ5	934.50 ± 4.10	232.13 ± 1.09	60.20%	80:20
JSHSJ4	902.50 ± 9.90	245.38 ± 0.11	57.25%	78:22
JSHSJ3	781.25 ± 0.78	163.13 ± 0.68	65.45%	83:17
XJCT2010	1296.49 ± 1.38	239.06 ± 4.89	68.86%	84:16
XJCT2011	1830.78 ± 9.39	277.89 ± 1.65	73.64%	87:13
XJCT 2012	1547.23 ± 3.04	246.89 ± 4.60	72.48%	86:14
XJCT 2013	995.24 ± 1.38	85.23 ± 3.83	84.22%	92:8
XJCT 2014	1891.19 ± 4.45	273.68 ± 5.49	74.72%	87:13
XJCT 2015	1985.58 ± 11.34	300.89 ± 6.36	73.68%	87:13
XJCT 2016	496.49 ± 9.98	41.12 ± 1.27	84.70%	92:8
XJCT 2017	371.77 ± 12.62	40.30 ± 0.99	80.44%	90:10
XJCT2018	1374.47 ± 7.73	180.61 ± 4.36	76.77%	88:12
XJCT2019	1395.83 ± 14.52	128.67 ± 5.29	83.12%	92:8
XJCT2020	1436.31 ± 11.25	224.72 ± 7.69	72.94%	86:14
STB (strong aroma-type Baijiu)				
LZLJTOUQU	483.60 ± 12.63	130.08 ± 1.08	57.61%	79:21
LZLJTQ1	449.23 ± 0.55	128.18 ± 0.57	55.60%	78:22
LZLJTQ2	356.10 ± 1.48	157.90 ± 3.06	38.56%	69:31
LZLJER	—	235.50 ± 0.51	—	—
MZDQ	331.75 ± 2.19	—	—	—
JNC	478.13 ± 11.38	27.11 ± 3.23	89.27%	95:5
WLY	407.70 ± 0.45	35.02 ± 1.79	84.18%	92:8
GJ1573	764.67 ± 1.85	268.99 ± 0.29	47.95%	74:26
SJF	695.54 ± 7.93	228.43 ± 2.40	50.56%	74:26
LZLJ2012	370.00 ± 2.12	—	—	—
LZLJ2013	389.83 ± 0.73	—	—	—
LZLJ2014	933.88 ± 8.87	661.25 ± 0.21	17.09%	59:41
LZLJ2015	1048.00 ± 11.46	939.83 ± 0.23	5.44%	54:46
LZLJ2016	708.90 ± 1.81	—	—	—
LZLJ2017	900.70 ± 1.07	213.35 ± 0.24	61.70%	81:19
LZLJ2018	932.63 ± 6.97	772.00 ± 2.12	9.42%	56:44
LZLJ2019	960.98 ± 0.13	926.00 ± 2.26	1.85%	51:49
LTB (light aroma-type Baijiu)				
FPLJ	—	135.28 ± 2.79	—	—
FJZC1988	—	558.33 ± 3.06	—	—
LBFJ	760.90 ± 9.45	43.76 ± 0.99	89.12%	95:5
FJQH20	576.55 ± 13.51	31.81 ± 1.27	89.54%	95:5
FJQXMR	240.80 ± 7.23	43.27 ± 0.99	69.54%	85:15
FJBF	234.42 ± 9.45	45.99 ± 2.39	67.20%	84:16

Note: —: Not detected in the sample. The samples were mainly divided into soy sauce aroma-type Baijiu (SSB), strong aroma-type Baijiu (STB), and light aroma-type Baijiu (LTB) in Table 2. The code of each Baijiu sample was composed of the Chinese initials of each Baijiu sample.

**Table 3 foods-11-02607-t003:** Mean concentration and proportion of lactic acid isomers in different aromatic Baijiu.

		Average Concentration ± Standard Deviation (mg/L)		
Category	Number of Samples	D	L	D/L
Soy sauce aroma-type				
Commercial Baijiu products	26	1300.18 ± 447.40	397.54 ± 289.81	78:22 ± 16.16
JSHSJ Vintage Baijiu (5–41)	10	1016.68 ± 281.63	308.35 ± 127.07	76:24 ± 4.26
XJCTJ Vintage Baijiu (1–11)	11	1329.22 ± 527.87	185.37 ± 96.46	88:12 ± 2.80
Strong aroma-type				
Commercial Baijiu products	9	495.84 ± 155.38	151.40 ± 98.26	80:20 ± 9.72
LZLJ Vintage Baijiu (2–9)	8	780.62 ± 264.98	702.49 ± 427.18	60:40 ± 11.99
Light aroma-type				
Commercial Baijiu products	6	453.17 ± 260.05	143.07 ± 206.94	90:10 ± 6.08

**Table 4 foods-11-02607-t004:** Recognition thresholds and flavor characteristics of lactic acid enantiomers.

Compounds	Recognition Threshold (mg/L)	Flavor Characteristics
D-lactic acid	194.18	sour taste
L-lactic acid	98.19	sour taste

**Table 5 foods-11-02607-t005:** TAVs of lactate enantiomers in the samples.

Sample	D-Lactic Acid	L-Lactic Acid
SSB (soy sauce aroma-type Bai-jiu)		
LM	8	2
JSHSJ1	4	1
JSHSJ2	4	2
JSJ	6	2
ZJ1	10	3
ZJ2	11	5
DYT1	6	2
DYT2	8	4
GT1	6	3
GT2	6	4
GT3	6	3
XJYZ	6	2
QHL	8	5
MT43	9	2
MT53	9	3
XJ1988	7	3
QJ1H1	8	4
DYT3	7	2
MTWZJ	8	3
LJ	9	8
MTCX	2	9
QJ1H2	9	5
TCSP	2	10
GZJSJ1	5	2
GZJSJ2	5	2
JSHSJ39	6	5
JSHSJ34	5	4
JSHSJ23	8	5
JSHSJ17	7	4
JSHSJ15	6	3
JSHSJ9	3	2
JSHSJ6	4	2
JSHSJ5	5	2
JSHSJ4	5	2
JSHSJ3	4	2
XJCT 2010	7	2
XJCT 2011	9	3
XJCT 2012	8	3
XJCT 2013	5	1
XJCT 2014	10	3
XJCT 2015	10	3
XJCT 2016	3	<1
XJCT 2017	2	<1
XJCT2018	7	2
XJCT2019	7	1
XJCT2020	7	2
STB (strong aroma-type Baijiu)		
LZLJTOUQU	2	1
LZLJTQ1	2	1
LZLJTQ2	2	2
LZLJER	—	2
MZDQ	2	—
JNC	2	—
WLY	2	—
GJ1573	4	3
SJF	4	2
LZLJ2012	2	—
LZLJ2013	2	—
LZLJ2014	5	7
LZLJ2015	5	10
LZLJ2016	4	—
LZLJ2017	5	2
LZLJ2018	5	8
LZLJ2019	5	9
LTB **(light aroma-type Baijiu)**		
FPLJ	—	1
FJZC1988	—	6
LBFJ	4	<1
FJQH20	3	<1
FJQXMR	1	<1
FJBF	1	<1

## Data Availability

The data presented in this study are available on request from the corresponding author.

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
