# Peer review of "Distribution and Quantification of Lactic Acid Enantiomers in Baijiu"

_foods, 2022, doi:10.3390/foods11172607_

Round 1
Reviewer 1 Report
The manuscript is somewhat confusing, especially regarding the samples analyzed, the nomenclatures used and what is the difference between each of the samples. I recommend clearly defining the samples analyzed in the manuscript, what each of the codes used means and what are the differences between the samples, for example in figure 2, table 2 it is necessary to define the samples in the footnotes to the figure and table .
The results described from line 231 are given in ranges as an example "The concentration range of D-lactic acid in SSB was 306.75 ± 6.11-1985.58 ± 11.34 231 mg/L, followed by STB (331.75 ± 2.19-1048.00 ± 11.46 mg/ L), it is not clear why this range is due? How many repetitions per sample were analyzed? What is the reason for not presenting an average value?
In general, the presentation of the results is not clearly arranged in order to understand the factors that differentiate the samples and from there to draw conclusions.
Reviewer 2 Report
I want to thank you for the opportunity to participate in the manuscript review.
The introduction to the article is written clearly, it introduces the topic well, the selection of literature is appropriate.
The purpose and scope of the article have been correctly defined. The methodology is clearly written and makes it possible to repeat the experiments. The results and discussion are well presented.
I consider the entire manuscript interesting and worthy of attention. The manuscript was pleasant to read.
Below are some minor notes on the manuscript.
Line 10-40. The abstract should be corrected according to the Guide and shortened.
Line 41. Please change some keywords so that they do not repeat with those appearing in the title of the manuscript. This will increase the possibilities of searching for an article in the database.
Line 58 - Please change the "R-enatiomer" to "R-enantiomer".
Line 71 - Please complete the space in the phrase "the Baijiu thick[12]."
Line 124 - The sentence should not start with a digit.
Line 125-126, 138 - Please add more information about the filtration performed.
Line 130 - Did the authors mean Chirex 3126 (D) -penicillamine?
Line 150-178 - Do the authors of the manuscript have permission from the Bioethics Committee to conduct experiments on humans?
Line 156 - Please complete the space in the phrase “(guidelines for determination of flavor thresholds of Baijiu)[25,26] .”
Table 1 - Please complete the space in the phrase "Linearity(mg/L)"
Line 262 - Please complete the phrase "acillus coagulans" (missing “B” - Bacillus coagulans)
Figure 5 - Please enlarge the charts to the end of the margins, they will be clearer. Please sign the OY axis (mg/L of what?)
Figure 6 - Please sign the OY axis (mg/L of what?). The 6c also lacks a unit.
Table 5 - Please try to fit the results in the table on one page.
Author Contributions - Please replace full names with the initials of the authors of the manuscript.
Reviewer 3 Report
The subject and scope of the manuscript "Distribution and quantification of lactic acid enantiomers in Baijiu" submitted for review is consistent with the subject of the Foods journal.
The manuscript needs to be refined.
First of all, the division (categorization) of the material and the description of it in subsection 2.2 seem unclear.
I also have comments on the construction of Tables 2 and 5, which seems to be chaotic. Also, the arrangement of Tables 2 and 3 in the text is incorrect.
A few errors also apply to the naming of microorganisms (lines: 262, 274).
It seems to me that the text requires linguistic and stylistic correction (numerous repetitions of the text).
Lack of interpretation of the results presented in subsection 3.6 (lines 362-372).
Note to the chapter "Conclusions" - too extensive, they cite the results once again, there is no general conclusion from the research carried out - at least in the light of the information provided in "Introduction" (lines 67-68).
All comments (also minor, not mentioned above) were introduced in the review mode to the submitted pdf file.

Reviewer 4 Report
This is an interesting paper on lactic acid enantiomers in Baiju. A few areas need to be checked: Where is the research question? Storage/aging time in years is a key factor - it needs to be accounted for in the results and in stats analysis. The sensory part needs to be revised. What type of sensory evaluation was used - was this a focus group? Any replications of the sensory evaluations? Where is the Human Ethics # for the sensory part?
Section on data analysis needs to be clearer: t-test is used to analyse data and not assess data? Data were analysed by MS Office tools and not organised by MS Office?
Use of detection limits needs to be clear.
Tables/Figures need to indicate number of replications
What is (-) in Tables? add a note.
Lines 222-247 need to be fully edited. Data in Tables should be discussed in the R and D, and not be repeated in text - only highlights. This also applies to lines 362-372.
Line 262 - taxonomy of microorganisms must be written accordingly
Conclusion needs to be overhauled. Only clearly formulated outcomes that answer the research question need to be included here. Do not repeat the discussion of results in the Conclusion.
Round 2
Reviewer 1 Report
The manuscript was substantially improved, providing greater clarity, especially in the information regarding the samples analyzed. Acceptance is recommended.